# Tracking the financial flows of Indonesia's COVID-19 vaccination program

**Olivia Herlinda**[1], **Ayudina Larasanti**[1]\*, **Sinta Amalia Kusumastuti Sumulyo**[1], **Sayyid Muhammad Jundullah**[1], **Indra Yudha Mambea**[2]

**1** Center for Indonesia's Strategic Development Initiatives, Research and Policy, Jakarta, Indonesia,
**2** Center for Economics and Development Studies, Universitas Padjadjaran, Bandung, Indonesia

\* ayudinalarasanti99@gmail.com, ayudina.larasanti@cisdi.org

## Abstract

Despite available funding mechanisms for COVID-19 vaccination programs, disparities persisted in certain areas. This study aimed to track the sources, allocation, and utilization of provincial and district-level government expenditures on the COVID-19 vaccination program in Indonesia to identify financial gaps affecting vaccination coverage. This study used a mixed-method approach to track the expenditure of Indonesia's COVID-19 vaccination program in 2021 and 2022 using the System of Health Accounts (SHA) framework. We collected expenditure data and conducted focus group discussions and in-depth interviews with government representatives from targeted provinces and districts. Case studies were conducted in four Provincial Health Offices (PHOs) and four District Health Offices (DHOs) in Indonesia. The results of expenditure tracking show significant variation in the expenditure for the COVID-19 vaccination program across subnational levels, based on factors such as fiscal capacity, political commitment, national and regional priorities, access challenges, geography, existing immunization infrastructure, and private sector engagement. Despite these variations, most of the programs were primarily funded by the National or Regional Budget, with subnational governments highly dependent on central government budget transfers. The largest expenditures generally included funding for the cold chain, distribution, vaccinator incentives, and per diem.This study found that Indonesia's complex health financing mechanism, coupled with limited fiscal capacity, struggles to ensure equitable vaccination delivery, especially in underserved areas. The budget transfer process from central to subnational levels is inadequate to account for vulnerabilities like geographical challenges. These disparities underscore the need for more coordinated and flexible health financing mechanisms during pandemics. Developing resource allocation guidelines and improving national-to-local resource distribution during health crises are essential for better outcomes.

**Data availability statement:** All data underlying the findings reported in the manuscript, have been provided as part of the submitted article.

**Funding:** The author(s) received no specific funding for this work.

**Competing interests:** The authors have declared that no competing interests exist.

## Introduction

Globally, declining COVID-19 transmission and mortality rates prompted the World Health Organization (WHO) to rescind the disease's designation as a Public Health Emergency of International Concern (PHEIC) on May 5, 2023. Despite this, COVID-19 continues to pose significant health risks, particularly to vulnerable populations. Disparities in vaccine access still persist worldwide. As of mid-2023, the proportion of the population that had received at least one dose of COVID-19 vaccine, reached 67.73% in lower-middle-income countries (LMICs) and 33.8% in low-income countries (LICs), markedly lower than the 80.27% and 80.77% observed in high-income and upper-middle-income countries (UMICs) respectively [1]. While supply constraints still exist, challenges in vaccine distribution within low-income and LMICs stem from fragile health systems [2], overburdened health workers [3], prioritization and resource allocation, and inadequate infrastructure [4], complicating efforts to achieve vaccine equity [5].

The Government of Indonesia (GoI) has made numerous efforts to reduce the number of positive cases and deaths caused by COVID-19 through the national vaccination program started in 2021 [6–8]. As outlined in the Technical Guidelines for the Implementation of Vaccination (Regulation No. HK.01.07/MENKES/4638/2021), in the context of Indonesia, the term "complete vaccination" refers to the completion of the primary vaccination series, which generally consists of two doses for most COVID-19 vaccines [9]. By 2023, the vaccination program has been operational for nearly two years, and the government had transitioned to the 'recovery' phase, aiming to manage endemic states and prevent future pandemics [10,11]. However, inequity in COVID-19 vaccine access remains a significant challenge. By mid-2023, although national primary dose coverage reached 70.49%, 18 provinces had not yet met the 70% target. Four provinces; West Sulawesi, Maluku, West Papua, and Papua, had coverage below 50%, with Papua being the lowest at just 23.38%. Coverage among specific groups was also limited; in West Kalimantan, East Nusa Tenggara, Central Sulawesi, and Southeast Sulawesi, less than 50% of the older population received the primary dose [12]. Additionally, coverage rates for other vulnerable groups remain unknown due to the lack of disaggregated data [13,14]. In this study, we define vulnerable groups as individuals who face significant barriers that increase their risk of COVID-19 infection and/or the severity of the disease. These barriers include systemic marginalization or discrimination, low health literacy, poor initial health status, limited access to public healthcare services, and other factors that contribute to their vulnerability [13]. Disparities occurred due to inadequate cold chain system, limited health worker resources, limited planning and fiscal capacity [15].

Historically, Indonesia has allocated relatively low expenditure on health, with only 2.69% of its Gross Domestic Product (GDP) in 2022. Most of this expenditure has been directed toward curative rather than preventive measures [16]. This low spending impacted both vaccine procurement and infrastructure development required for equitable distribution [17]. However, effective planning, priority setting, and resource allocation remains imperative in resource-constraint settings to ensure resources are utilized optimally to obtain maximum impacts [18].

During the COVID-19 pandemic, many local governments in Indonesia repurposed existing resources to finance COVID-19 measures. Nonetheless, there have been limited financial evaluations of COVID-19 programs, including studies assessing the expenditures of the COVID-19 vaccination program. Tracking expenditures and disbursements, and ensuring effective utilization, remains challenging given the varied capacity of subnational governments [19]. Some studies have evaluated the costs of routine immunization outreach during the pandemic and the cost of treating COVID-19 patients [20,21]. Moreover, economic evaluations of vaccination programs typically use cost-effectiveness analysis or budget impact analysis, primarily for new vaccine introductions [22–24]. For example, a study in Colombia comparing two COVID-19 vaccination strategies found that prioritizing high-risk groups was consistently more cost-effective in low-resource settings, while a study in South Africa's West Rand district, which estimated delivery costs across multiple channels, identified vaccine procurement and staff time as key cost drivers, highlighting the importance of detailed expenditure tracking to understand vaccination program finances [25,26]. Despite such insights, studies using resource and expenditure tracking methods for COVID-19 vaccination programs remain limited and tend to focus more on general health services, specific basic health services, and tuberculosis (TB) programs at the subnational level [27–29]. To address the gap in studies on COVID-19 vaccination program financing, this study aimed to track the sources, allocation, and utilization of government expenditure for the COVID-19 vaccination program at the subnational level to address the gap in financing studies. The findings are intended to improve monitoring, coordination, and accountability, while offering recommendations for more effective resource allocation in future pandemics.

## Methods

### Ethics statement

This study has been approved by Research Ethics Committee of Universitas Katolik Indonesia Atma Jaya number 0002T/III/PPPE.PM.10.05/03/2023. All participants provided written consent to participate in this study through google form.

### Study setting

This study investigated the financing of government-funded COVID-19 vaccination programs in Indonesia. In addition to examining national-level policies, this study explored vaccine financing at the subnational level to capture detailed expenditures, given the roles of Provincial Health Offices (PHOs) and District Health Offices (DHOs) as frontliners in national vaccine delivery. The study focused on COVID-19 vaccine-related expenditure at the provincial and district levels; routine immunization spending data were not collected. Publicly available subnational data on routine immunization were sought but were unavailable and lacked sufficient detail to disaggregate different kinds of expenditures. The recruitment and data collection period for this study was conducted from March 28 to June 30, 2023.

### Study population

The health offices included in this study (see Table 1) were purposively selected based on varying levels of primary dose vaccination coverage in 2023 to reflect different regional situations and characteristics. The chosen provinces were categorized into three groups: provinces with primary dose vaccination coverage over 70%, between 50–70%, and less than 50%. This classification aligns with the WHO global's and Indonesian government's target of 70% COVID-19 primary dose vaccination coverage [30,31]. Among the provinces that met the vaccination coverage criteria, specific provinces were purposively selected based on the feasibility of data collection and geographic diversity. The selected provinces also represent different islands of Indonesia, supporting a more comprehensive understanding of regional variation. Within each selected province, one district was purposively chosen to capture intra-provincial variation and gather diverse perspectives from DHOs. District selection considered not only vaccination coverage but also demographic and geographic characteristics. For instance, Sigi, Seram Bagian Barat, and parts of Tanggamus face accessibility challenges due to remote and mountainous terrains, while Gianyar represents a more urbanized district with stronger healthcare

**Table 1. Characteristics of the regions studied.**

| Regions | Area (km sq) | Population (thousand) | Human Development Index | Population in Poverty (%) | Gross Regional Domestic Product (trillion rupiah) | Number of Primary Health Centers (Puskesmas) | Covid-19 Vaccination Coverage |
|---|---|---|---|---|---|---|---|
| **Province** | | | | | | | |
| Bali | 5,590 | 4,317,404 | 77.10 | 4.25 | 245.23 | 120 | 97.9% |
| Central Sulawesi | 61,606 | 3,066,143 | 70.95 | 12.41 | 323.62 | 215 | 53.9% |
| Lampung | 33,575 | 9,176,546 | 71.15 | 11.11 | 414.13 | 315 | 66.9% |
| Maluku | 46,914 | 1,881,727 | 70.94 | 16.42 | 14.14 | 222 | 45.0% |
| **District** | | | | | | | |
| Gianyar (Bali) | 364 | 515,344 | 79.24 | 4.47 | 27.94 | 13 | 90.4% |
| Sigi (Central Sulawesi) | 5,225 | 266,812 | 69.93 | 12.83 | 10.6 | 19 | 37.1% |
| Tanggamus (Lampung) | 2,902 | 652,898 | 67.86 | 10.5 | 18.12 | 26 | 60.2% |
| Seram Bagian Barat (Maluku) | 6,948 | 217,958 | 65.83 | 22.39 | 3.5 | 17 | 24.5% |

Sources: [33–35].

infrastructure. These varied contexts provided a solid foundation for comparing COVID-19 vaccine financial flows across regions. Initially, two provinces and two districts were selected for each vaccination coverage category; however, Jakarta and Papua were later excluded due to incomplete expenditure data. A map of the regions (Fig 1) and a table summarizing their demographic characteristics (Table 1) are provided below.

## Study design

This study used a mixed-methods approach and a case study design for an in-depth exploration of the System Health Accounts (SHA) framework.

**Quantitative.** This study employed an expenditure tracking approach to identify financing gaps, resource needs, optimize delivery strategies, and inform funding planning for vaccination programs [36]. We tracked expenditures for Indonesia's COVID-19 vaccination program expenditures in 2021 and 2022, focusing on this period as the program

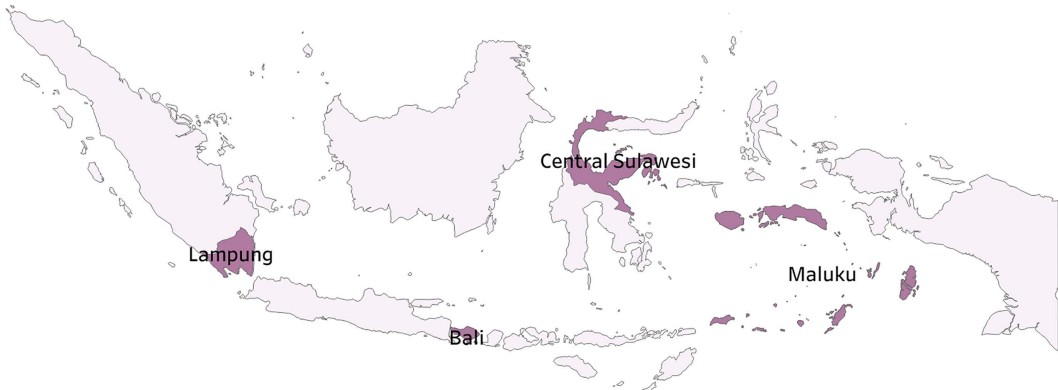

**Fig 1. Map of Indonesia (Created in Tableau using data from Natural Earth (public domain):** http://www.naturalearthdata.com**) [32].**

expanded to subnational level in 2021. The 2020 budget was available only at the national level, covering vaccine procurement.. Expenditure data were collected from selected provinces and districts using a specialised format, modified from the existing DHO format, to address regional variations in financial reporting. Additional data were gathered through in-depth interviews with key informants from health offices and a review of program documentation for validation. Despite the structured format and interviews, some health offices could not provide detailed expenditure information, creating challenges in comparing quantitative data. To provide the robustness of the findings, data triangulation was employed, integrating interviews, official reports, and program documentation. Normalization was achieved by aligning cost categories and standardizing expenditure components across provinces and districts to ensure greater consistency and comparability. In cases of scarce or inconsistent data, these measures enabled a more reliable comparison of expenditures. The microcosting analysis was finalized by excluding Papua and Jakarta due to insufficient data, resulting in a final dataset from four provinces and four districts, allowing for a more accurate and consistent comparison of expenditures.

All expenditures were converted into US dollars using the 2021 exchange rate, with US$ 1 was equivalent to IDR 14,308 [37]. This conversion was conducted to enhance international comprehensibility and enable cross-country comparisons. A constant exchange rate was applied throughout the two-year analysis period to ensure methodological consistency and to avoid distortions arising from exchange rate fluctuations. This approach provides a more stable and comparable basis for assessing financial performance over time. Official statistics indicate that health-related inflation was relatively modest (1.68% in 2021 and 2.87% in 2022) [37], indicating a minimal impact on the financial results presented in this study, and thus, no further inflation adjustment was deemed necessary.

Micro-costing is the preferred method for calculating health intervention costs, as it measures specific resource consumption and unit costs. Due to the unavailability of some specific data, we employed an expert judgement approach to estimate actual expenditure. Experts were asked to estimate the percentage use of resources including supplies, personnel, equipment, transport, and other relevant inputs for each service component. These estimates were provided by individuals with operational knowledge of the service delivery process [38–40]. We also collected COVID-19 vaccination coverage data as of June 2023 from the MoH's vaccine dashboard [12]. Then we calculated total expenditure per capita by dividing the COVID-19 vaccination expenditures for 2021 and 2022 by the total target vaccination population.

**Qualitative.** From March to June 2023, we conducted three Focus Group Discussions (FGDs) with 24 key informants from six ministries and government agencies, four PHOs, four DHOs, and six civil society organizations (CSOs).. To delve deeper, we conducted eight in-depth interviews with four informants from PHOs and four from DHOs, who also participated in the FGDs. The inclusion criteria for informants in both processes were at least six months of work experience, preferably in finance, planning, or program departments, with knowledge of COVID-19 vaccination program financing and willingness to be recorded. Informants who did not meet these criteria were excluded. The qualitative study aimed to triangulate and contextualize findings from the quantitative data, and explore implementation challenges in the COVID-19 vaccination program, particularly related to financing and fund management. Triangulation focused on validating budgeting processes, resource allocation, and coordination across levels.

## Analysis

Indonesia has developed health accounts at the national (National Health Account/NHA), provincial (Provincial Health Account/PHA), and district levels (District Health Account/DHA) [41]. This study's evaluation is based on a health system perspective and uses a conceptual framework of PHA and DHA derived from the WHO System of Health Accounts (SHA) theoretical framework (2011) to classify costs from expenditures. The SHA-11 framework systematically maps health service financing flows, offering a comprehensive overview of healthcare expenditure [41,42]. The framework has been applied in a study across 36 LMICs to track primary healthcare expenditures [43], and in the immunization program in Honduras, where it identified financial gaps in cold chain expansion, social mobilization, information systems, training, and research [44].

This study adopts the SHA-11 framework and employs three dimensions most relevant to our research objectives: Financial Source (FS), Health Care Function (HC), and Programme Type (PR). Following the 2021 USAID and MoH study, we incorporated a fourth dimension, Health Input (HI), to align with how the Indonesian government categorizes expenditures. Table 2 provides a brief description of each dimension, and S1 Appendix details the operational definitions of the variables utilized to classify the expenditure in this study.

## Reporting

For manuscript reporting, this study follows the Consolidated Health Economic Evaluation Reporting Standards (CHEERS) 2022 guidelines [45].

## Results

### 1. Trends in COVID-19 vaccination budget and expenditure

During the COVID-19 response from 2020 to 2022, the budget allocation for the vaccination program reflected the shifting priorities and intensity of the pandemic. In 2020, US$2.497 million was allocated, primarily for vaccine procurement. The highest allocation occurred in 2021, reaching US$5.171 million, driven by the national vaccine rollout and peak infection and mortality rates. [46] By 2022, the allocation decreased significantly to US$1.108 million, reflecting reduced overall COVID-19 funding amid more controlled transmission and declining public demand for vaccination. [47–49]

**Provincial level.** At the provincial level, Central Sulawesi and Maluku PHOs saw an increase in expenditure from 2021 to 2022, while Bali and Lampung PHOs experienced a decrease (see Fig 2a). In 2021, Lampung recorded the highest expenditure among the four provinces, primarily due to reallocations and support from the regional budget for medical equipment. This was driven by the need to establish cold chain infrastructure capable of maintaining temperatures below 8 degrees Celsius, as the province lacked an adequate system prior to the pandemic and had to support a large-scale vaccination program in remote areas. Lampung's larger population of 7,558,816 also contributed to its higher spending [12]. In 2022, its expenditure declined sharply as major investments in medical equipment, cold chain, and infrastructure had already been made in the previous year.

**District level.** Provincial and district-level expenditures were not always aligned. In 2022, while Lampung's provincial spending decreased, Tanggamus district's spending increased. A similar trend was observed between Sigi District and Central Sulawesi Province. Sigi DHO had the highest expenditure in 2021, mainly for medical equipment and cold chain infrastructure, but saw the largest decrease from 2021 to 2022 as procurement needs were fulfilled. Fig 2b shows that Tanggamus and Seram Bagian Barat (SBB) increased their expenditures from 2021 to 2022, while Gianyar and Sigi reduced theirs. Tanggamus's spending in 2022 rose tenfold to cover vaccine transportation costs and unpaid vaccinator incentives from 2021.

*"Budget from 2021 was only $24,462 spent; the remaining incentive was paid in 2022."(Representative of Tanggamus DHO)*

**Table 2. Description of expenditure tracking dimensions.**

| No | Dimension | Description |
|----|-----------|-------------|
| 1. | Financial Source (FS) | Classifies what sources of funds are received by each funding scheme. |
| 2. | Health Care (HC) Function | Classifies the goods and services consumed by individuals or populations, such as curative, rehabilitative, preventive, long-term, supportive, medical, governance, administration, and financing. |
| 3. | Programme (PR) Type | Captures whether expenditures are specific to a certain health program (COVID-19 Vaccination) or integrated with other health programs. |
| 4. | Health Input (HI) | The type of input purchased by service/program providers to carry out activities, including operational, maintenance, and investment expenditure. |

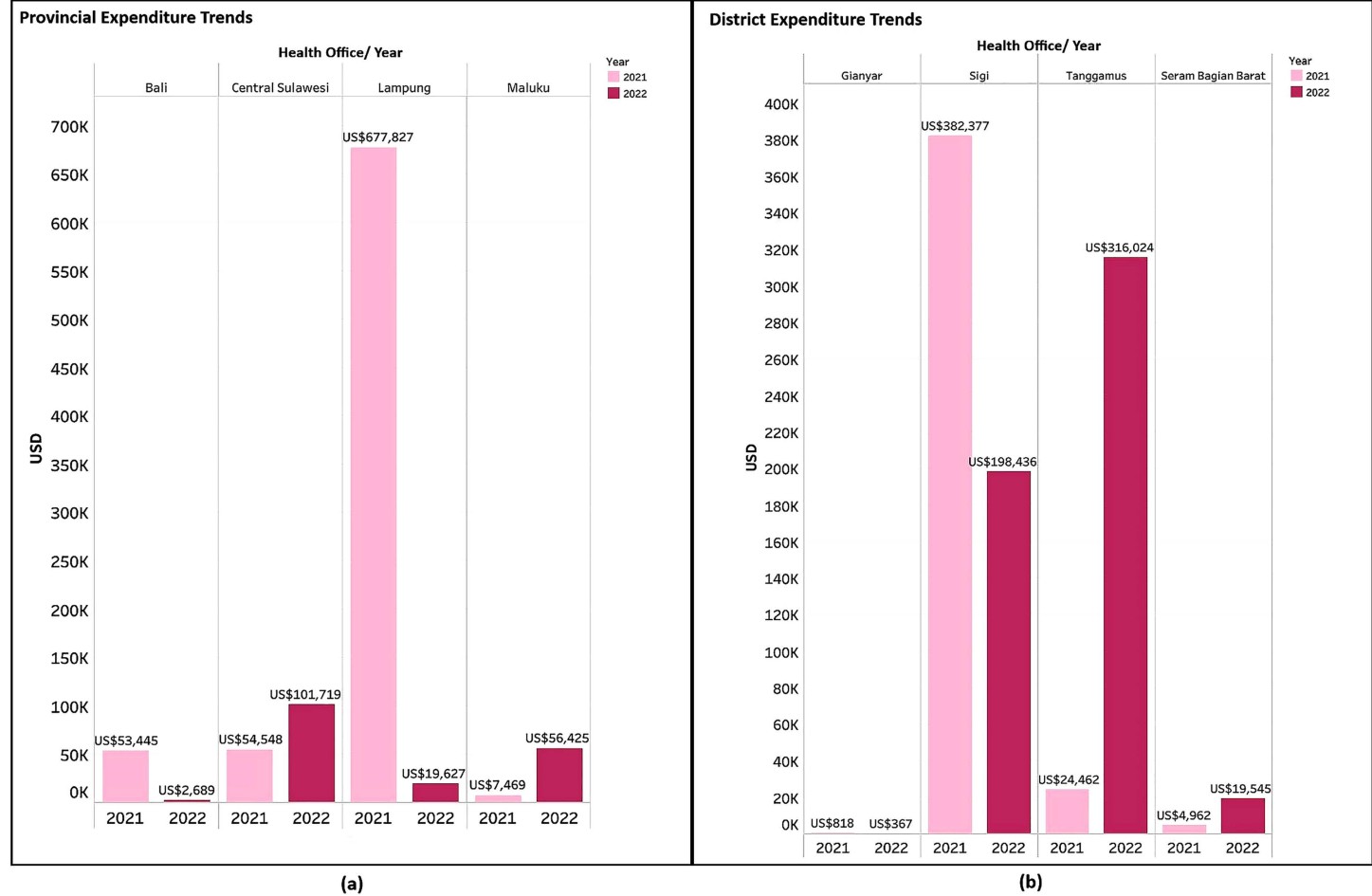

**Fig 2. Provincial and District Health Office Expenditure trends for the COVID-19 vaccination program 2021 and 2020.**

Gianyar had the lowest expenditure in both 2021 and 2022, as it did not allocate a specific budget for the COVID-19 vaccination program. Instead, funding for the vaccination program came from the budget for COVID-19 prevention, control, monitoring, and evaluation. Despite minimal funding, Bali Province, including Gianyar, benefited from substantial support through cross-sector collaborations with ministries and the private sector, due to its significance to national tourism.

## 2. Total expenditure of COVID-19 vaccination program compared with COVID-19 vaccination coverage

The Indonesian government set a target of achieving 70% COVID-19 vaccination coverage by the end of 2022 [50]. This section compares vaccination coverage across provinces and districts with their total vaccination expenditure per capita, calculated by combining 2021 and 2022 expenditures and dividing by the vaccine targets. Fig 3 shows coverage rates as of June 2023 [12].

Bali PHO had the lowest expenditure per capita (US$0.0148) in 2021–2022 but achieved 97.94% primary dose coverage, the highest among the provinces studied. Extensive private sector support, providing funds, meals, transport, and incentives for vaccinators, strengthened Bali's mass immunization efforts. This, combined with strong local government and tourism industry support, helped Bali achieve the second-highest coverage in Indonesia, after Jakarta [50].

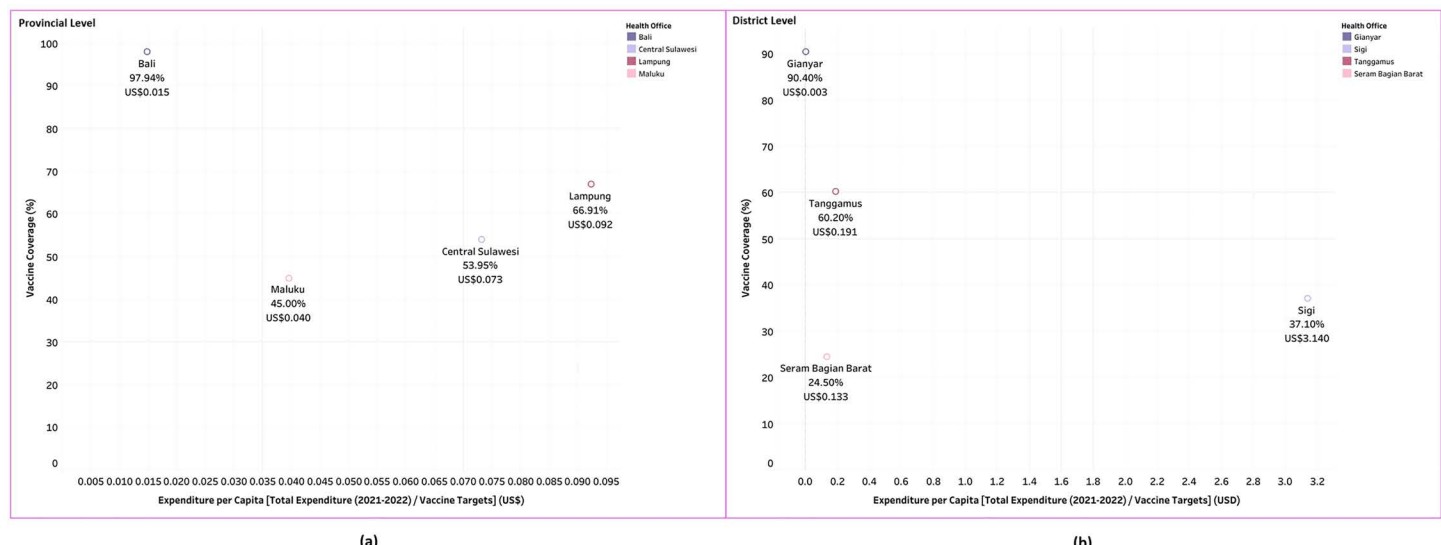

**Fig 3. Scatterplot of COVID-19 vaccination program expenditure per capita (USD) and vaccine coverage at the provincial and district Level.**

*"Our vaccination team is invited by the Central Bank, they pursue vaccine targets. Sometimes tourism agents too. As a vaccine team, we were given transportation, food and drinks like that. In 2021, we are more often invited, many malls invite us to carry out vaccinations, and we are facilitated for lunch, and that's how the private sector gets involved."* (Representative of Bali Provincial Health Office)

In contrast, Maluku had the lowest vaccination rate (45.03%) and the second-lowest expenditure per capita (US$0.0396), followed by Central Sulawesi (53.95%; US$0.073) and Lampung (66.91%; US$0.092). The data indicate a relatively linear relationship at the provincial level with minimal private sector involvement: higher per capita expenditure generally corresponds to higher vaccination coverage.

At the district level, Gianyar DHO, like Bali Province, achieved the highest vaccination coverage (90.4%) among DHOs. However, as shown in Fig 3b, higher expenditure did not consistently lead to higher coverage. For example, Sigi DHO had the highest per capita expenditure (US$3.14) but only reached 37.1% coverage, as most funds were spent on transportation and vaccine storage due to geographic challenges. Similarly, SBB DHO, with the lowest vaccination rate (24.5%), allocated most of its budget to distribution and transportation, reflecting the region's logistical difficulties. Both SBB and Sigi are classified as 3T areas (least developed, frontier, and outermost regions) due to limited economic development, infrastructure, human resources, financial capacity, and accessibility [51].

## 3. Expenditure tracking

The following section outlines COVID-19 vaccination program expenditures at the provincial and district levels, classified using the SHA-11 framework dimensions: Financial Source (FS), Health Care Function (HC), Program Type (PR), and Health Input (HI) budget line items. Key overall findings include:

**a. Financial Source (FS).** The COVID-19 vaccination program involved collaboration among the government, civil society organizations, and the private sector, with funding primarily from central and local government budgets. Fig 4 illustrates the flow of vaccination funds from the national to subnational levels, involving multiple government tiers and non-state actors. The central government's primary funding came from the state budget, supplemented by international

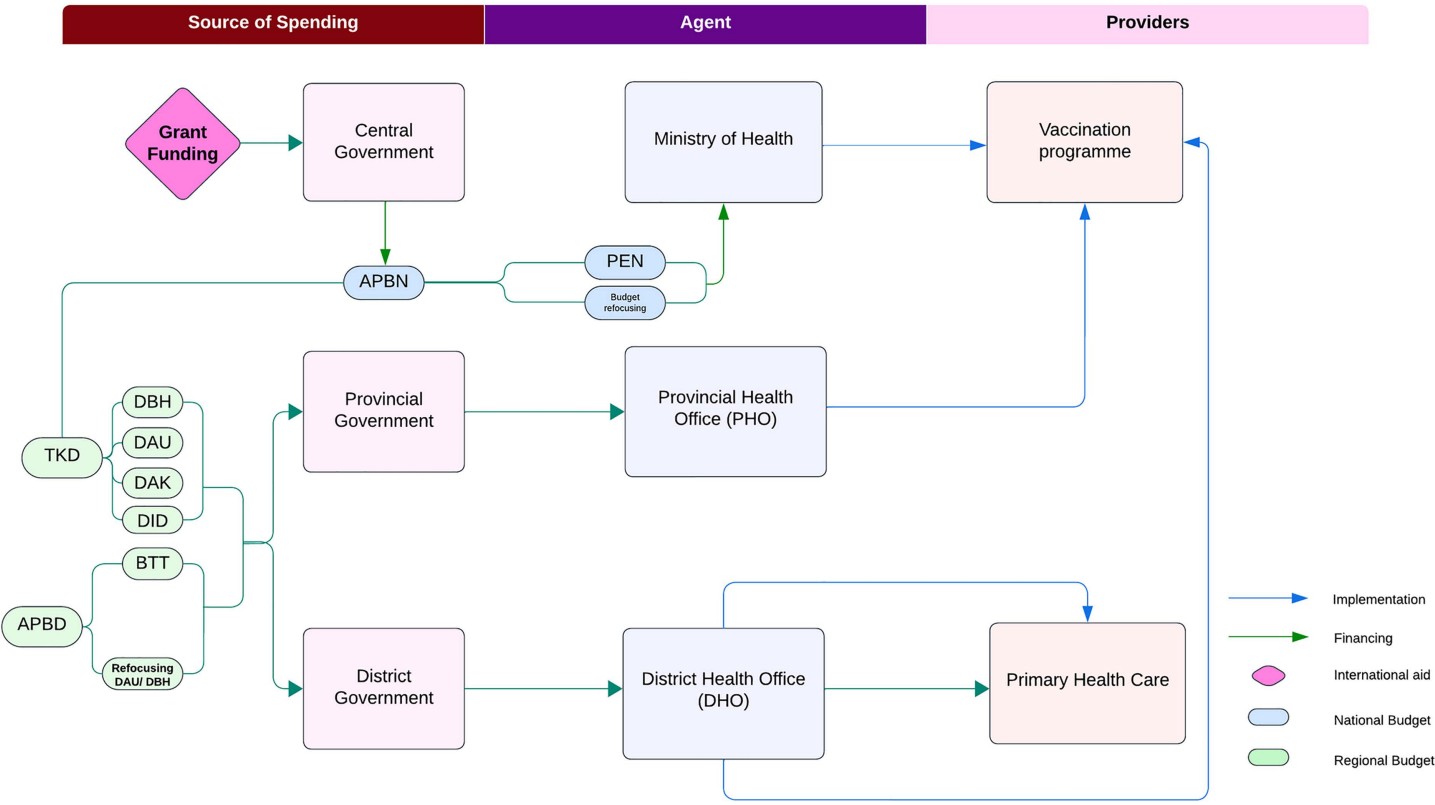

**Fig 4. The flow of funding for National COVID-19 Vaccination Programs in Indonesia. Source:** worldbank.org, 2016 [52]. **Note:** [a]APBD: *Anggaran Pendapatan dan Belanja Daerah* (Local Government Budget), [b]APBN: *Anggaran Pendapatan dan Belanja Nasional* (Central Government Budget), [c]BTT: *Belanja Tidak Terduga* (Unexpected Expenditure Funds), [d]DAK: *Dana Alokasi Khusus* (Specific Allocation Fund), [e]DAU: *Dana Alokasi Umum* (General Allocation Fund), [f]DBH: *Dana Bagi Hasil* (Revenue Sharing Fund), [g]DID: *Dana Insentif Daerah* (Regional Incentive Funds), [h]PEN: *Program Pemulihan Ekonomi Nasional* (National Economic Recovery Program), [i]TKD: *Transfer ke Daerah (*Transfer to Regions).

grants. These funds could be allocated directly to district governments, with allocations based on the district's fiscal capacity, national priorities, among other considerations.

At the subnational level, governments could use their own budgets (provincial or district APBD) or access reserve funds (BTT) for emergencies. However, in many cases, BTT funds were insufficient for pandemic response needs. Regions like Maluku PHO relied on BTT for COVID-19 vaccination expenses in both 2021 and 2022. These varied funding sources highlight the multiple financing options available to subnational health offices. However, the MoH's technical guidelines prohibit provincial and district governments from using multiple funding sources for the same activity (double costing) [53].

The central government primarily handled vaccine procurement and delivery to the provincial level, after which PHOs distributed them to DHOs using their own budgets. DHOs managed vaccine storage, distribution to health facilities, logistics, and healthcare worker incentives. With limited MoH support, PHOs and DHOs had to secure additional local resources to meet their needs, resulting in varying expenditure components based on local policies. Fig 5a shows that all PHOs relied on regional budgets from multiple sources for the COVID-19 vaccination program.

In 2021, Bali's regional budget for COVID-19 vaccination was sourced from the refocused General Allocation Fund (DAU), a transfer from the central government. This shift likely resulted from a decline in Bali's Regional Original Revenue (PAD) due to the tourism sector downturn caused by the COVID-19 pandemic [54,55]. In 2022, according to an informant, since no new regulations permitted the use of the DAU, funding for the vaccination program came from the reserve fund (BTT).

*"In 2022, no extended regulation allowed us to refocus the DAU to plan from the very beginning the same amount of budget as 2021; we do not have the authority, so the source of the budget is then taken from BTT." (Representative of the Bali Provincial Health Office)*

At the district level, Fig 5b shows that funding for COVID-19 vaccination programs came from both the national (APBN) and regional (APBD) budgets. Subnational governments could use multiple funding sources, but most districts relied heavily on the national budget, particularly through national budget - special allocation fund (APBN-BOK/ DAK) Non Physical, followed by regional incentive fund (APBN-DID). The variation in funding sources in 2022 reflected reduced budget allocations and shifting priorities for COVID-19, except for acceleration efforts in low-coverage areas like SBB. Only Sigi and Tanggamus allocated a portion of their vaccination funding from regional budgets.

In 2022, budget variations arose as DHOs required additional funds for vaccine acceleration programs, despite reductions in national budgets and COVID-19 priorities. To meet these needs, DHOs diversified their funding sources, utilizing reallocations from APBD-DAU, APBN-BOK/DAK non-physical, and APBN-DID. The reallocation of APBN-BOK/DAK non-physical was primarily used at the district level, reflecting their responsibility for managing primary healthcare facilities [56]. Meanwhile, the APBN-DID allocated for COVID-19 vaccination services was used by Tanggamus DHO to provide food and beverages during vaccination delivery.

Sigi DHO had the highest expenditure in 2021, drawing from regional budgets via the refocused DAU and Revenue Sharing Fund (DBH). In contrast, Gianyar had the lowest vaccine program expenditures, where it spent only US$818 in 2021 and US$367 in 2022 from the APBN-BOK/DAK non-physical allocated for COVID-19 monitoring and control. This limited funding impacted the availability of resources for vaccinators and for monitoring and evaluation activities.

*"In terms of budget, there was no specific allocation for the vaccination program from the start. In 2021–2022, we used the COVID-19 response, monitoring, and control budget for vaccine-related expenditures. Since Gianyar was a green*

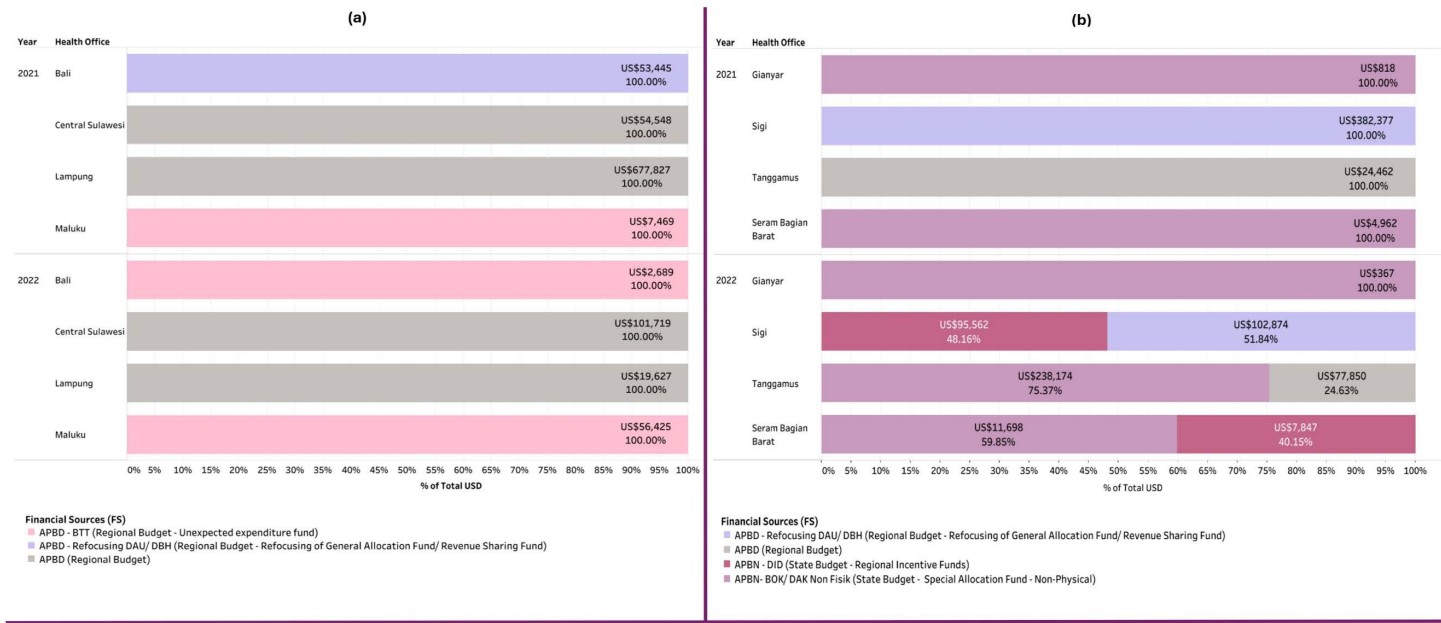

**Fig 5. Financial source at the provincial level and district in 2021 and 2022.**

*zone, we collaborated with community leaders, hospitals, and the private sector." (Representative of Gianyar District Health Office)*

**Non-state actors support:** The involvement of non-state actors in vaccination funding, such as providing meals and transportation for field teams, was not captured in this analysis, as these external parties managed their own financing. Vaccination efforts organized by external parties were typically funded in-kind, rather than through DHOs or PHOs.

*"No funding has been provided to us, but we are invited by the central bank, sometimes also by tourism actors. We carry out our role as a vaccine team, with transportation, food, and drink provided by them." (Representative of the Bali PHO)*

However, in some districts and provinces, non-state actors reported challenges in coordinating with state actors, limiting their ability to contribute additional financial resources to COVID-19 vaccination programs.

*"A lot of local governments fail to recognize the need for vaccinations, claiming that it is the community's responsibility. They take it lightly once vaccination targets are reached…. We tried to maximize our efforts, but then ran into policy barriers." (Representative of a CSO)*

**b. Health care function (HC).** The health care function (HC) of the COVID-19 vaccination program comprises public health and prevention services, medical equipment and materials, supporting services, and other health services. Public health and prevention services include costs associated with program implementation, such as meals during vaccination activities, travel expenses, and honoraria for vaccination teams. Medical equipment and materials cover pharmaceuticals, disposable medical supplies, and other medical devices. Supporting services encompass activities such as advocacy, meeting refreshments, and speaker honoraria. Other health services involve community empowerment initiatives and additional health-related activities. Subnational governments directed their budgets toward various program priorities, with significant shifts observed from 2021 to 2022. Most PHOs and DHOs prioritized public health and prevention services, focusing primarily on vaccination program implementation, followed by medical equipment and support services like monitoring and evaluation activities.

Fig 6a indicates that in both 2021 and 2022, Bali and Central Sulawesi dedicated the largest share of their expenditures to public health and prevention services. In contrast, Lampung and Maluku emphasized medical equipment and materials, with Lampung investing in cold chain facilities and Maluku focusing on vaccine distribution due to their geographical conditions which have more remote areas and islands. In 2022, Maluku allocated the highest budget proportion (49.55%) to support services for monitoring and evaluating vaccine storage at the district level.

At the district level, Fig 6b shows that SBB, Sigi, and Tanggamus DHOs allocated the largest portions of their budgets to public health and prevention services. These expenditures covered honorariums, transportation, meals for the vaccination team, and the procurement of screening forms and stationery, representing the largest budget share across most health offices. In contrast, Gianyar allocated its entire budget to supporting services for monitoring and evaluating COVID-19 prevention and control at Puskesmas, as it lacked a specific budget for the vaccination program.

**c. Programme type (PR).** Most PHOs and DHOs allocated specific expenditures to COVID-19 vaccination activities in 2021 and 2022, with many of these expenses dedicated solely to the vaccination program, separate from other initiatives. However, Lampung allocated a larger portion of its budget to integrated components due to cold chain infrastructure

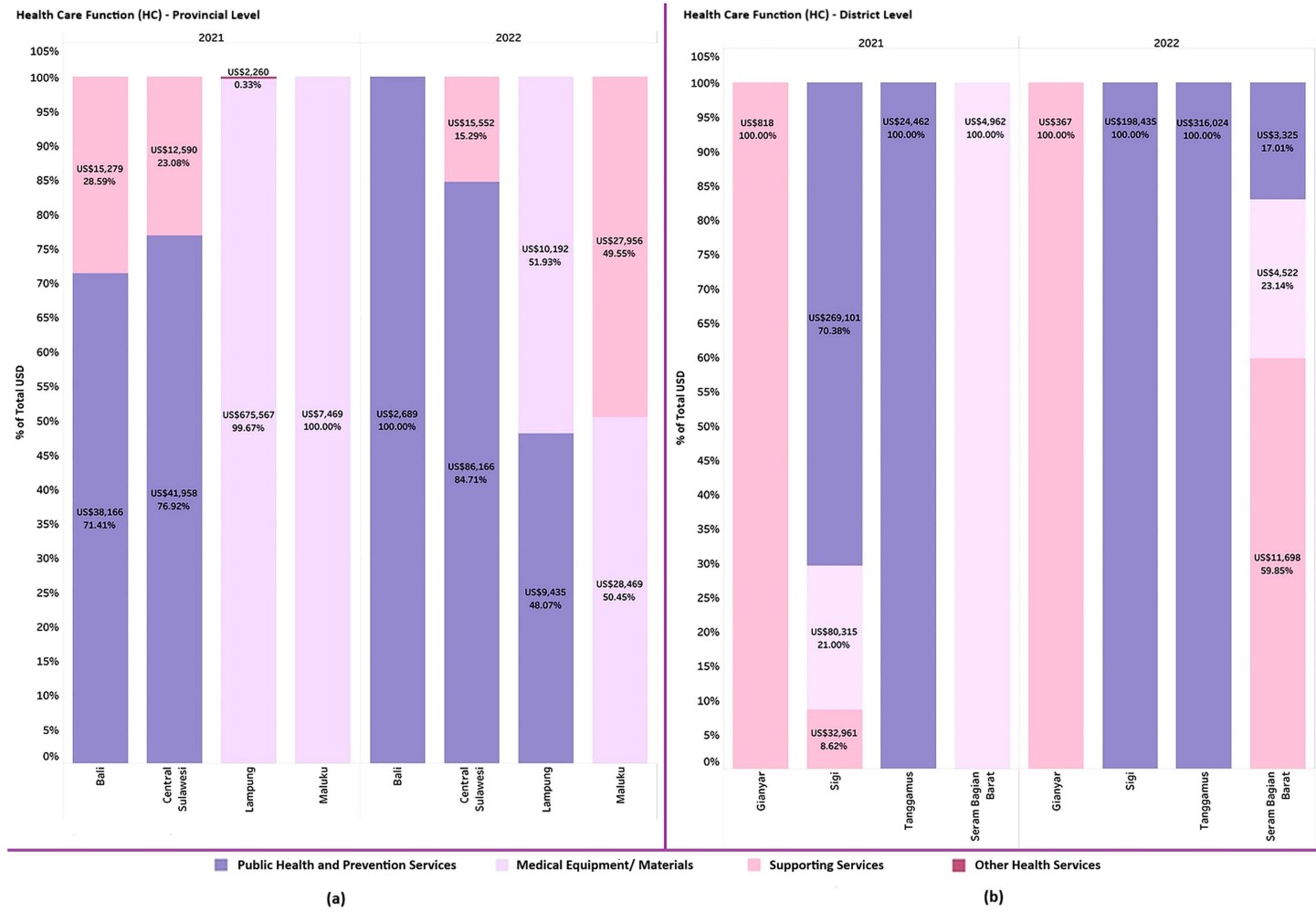

**Fig 6. Health care function by provincial and district level.**

procurement. Fig 7a shows that most provinces maintained distinct COVID-19 vaccination program funding, with Central Sulawesi and Maluku PHOs dedicating all their funds specifically to vaccination efforts. In contrast, Lampung PHO integrated its vaccination program into other health initiatives, especially in 2021. As the budget declined, the balance between integrated and specific funding became more even, driven by investments in cold chain facilities and medical waste management systems that supported hospitals, health centers, and pharmaceutical installations and could aid other programs.

At the district level (Fig 7b), most expenditures were specifically allocated to COVID-19 vaccination activities, except in Gianyar, which relied on integrated funding for both years.

*"From the start, there was no specific budget for vaccination. In 2021–2022, we used the monitoring and COVID-19 control budget... The central government did not provide a dedicated vaccination component." (Representative of Gianyar DHO)*

SBB and Tanggamus DHOs allocated specific expenditures for the vaccination program without integrating them with other activities in both years. In contrast, Sigi DHO combined integrated and specific expenditures. The proportion of

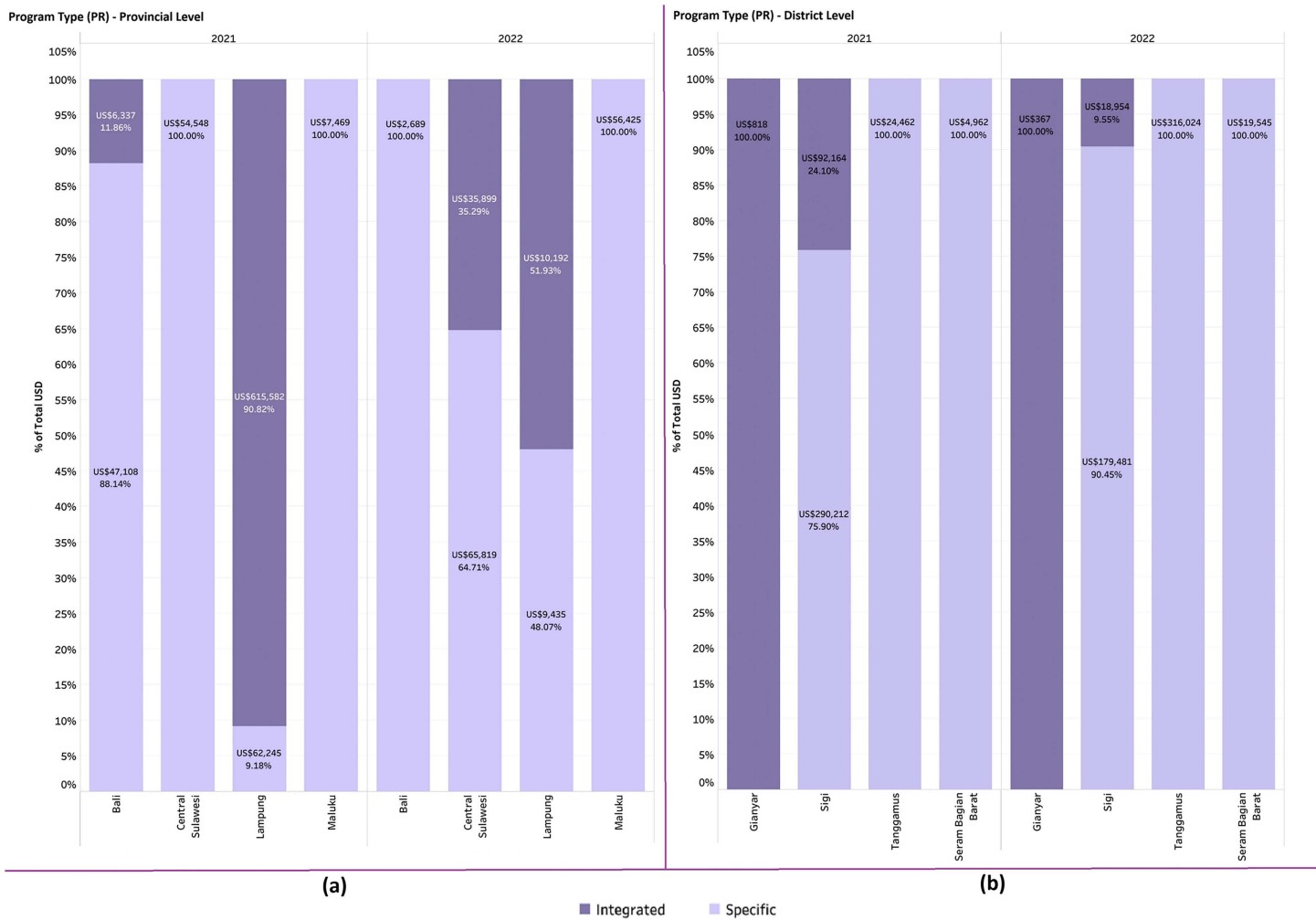

**Fig 7. Programme type by provincial level and district level.**

specific spending in Sigi increased from 75.9% in 2021 to 90.5% in 2022, as integrated expenditure dropped from 24.1% to 9.55%. In 2021, funds were needed for cold chains, including air conditioners, refrigerators, and electrical systems. By 2022, integrated spending was limited to increase the COVID-19 team's operational vehicles.

**d. Health input (HI).** Fig 8a shows that in 2021, three of the four PHOs; Bali, Central Sulawesi, and Maluku, allocated most of their expenditures to operational items, such as vaccination team incentives, vaccine distribution, and transportation costs. Lampung PHO was the exception, focusing more on investment due to cold chain procurement. By 2022, three out four PHOs shifted their budgets entirely to operational items, as previous investment needs, like cold chain infrastructure, had already been met. Only Maluku PHO has maintenance items for monitoring and evaluation.

At the district level, Fig 8b shows that in both 2021 and 2022, most expenditures were allocated to operational activities, except for Gianyar DHO, which spent exclusively on maintenance, particularly for monitoring and evaluation. In 2021, Sigi DHO was the only district to allocate funds across all three components: investment, operations, and maintenance.

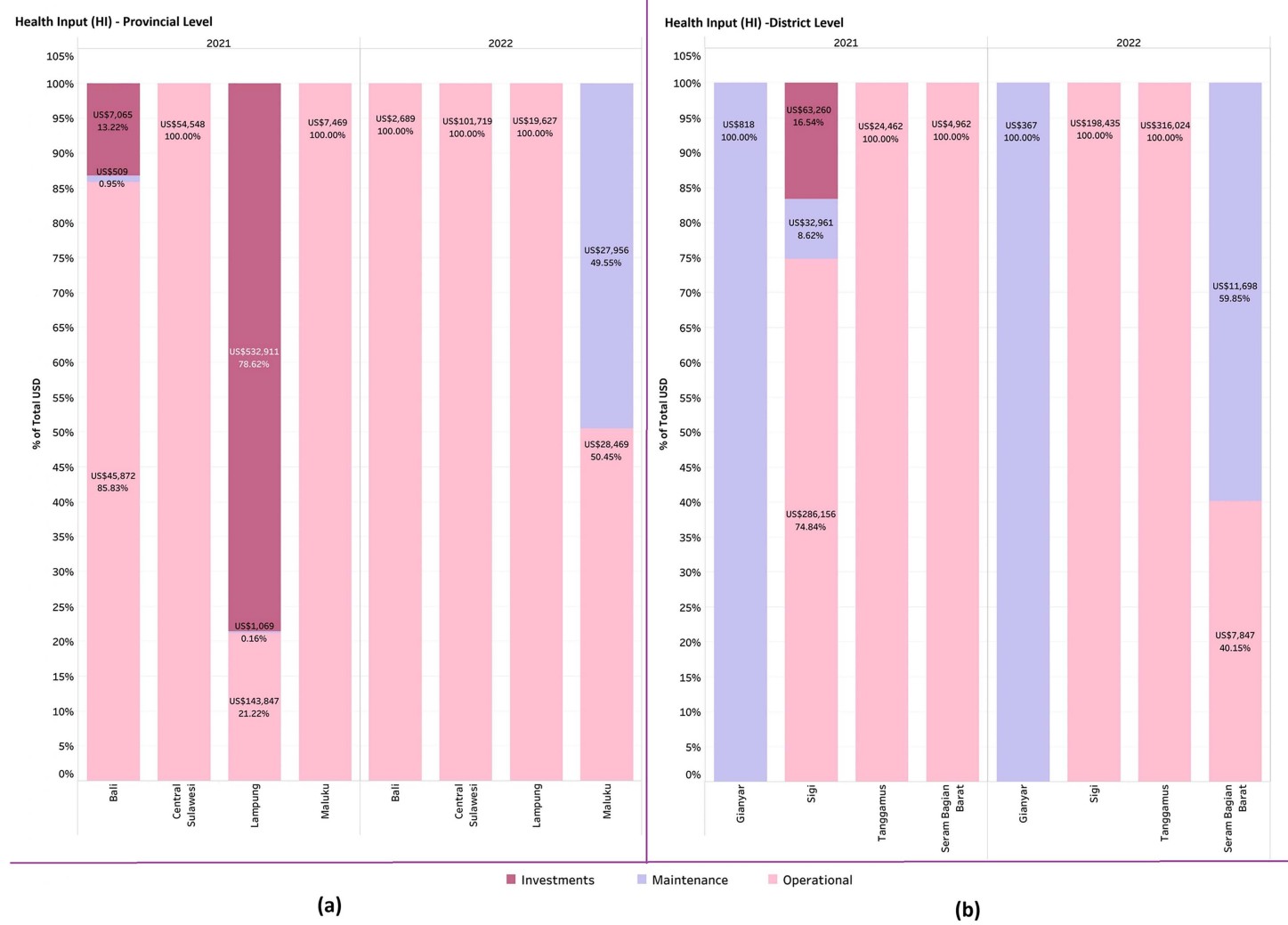

**Fig 8. Health input by provincial level and district level.**

The allocation of expenditure components for the COVID-19 vaccination program varied based on each region's capabilities and needs. As a result, some components available in one region were absent in others due to funding limitations. For example, not all regions allocated funds specifically for vaccine distribution.

*"It is true that we do not budget specifically for COVID-19 vaccine distribution; it is integrated with other medicines and medical equipment. So, when the pharmaceutical departments distribute medicines to districts, they distribute COVID-19 vaccines at the same time." (Representative of Central Sulawesi PHO)*

## Discussion

Indonesia's health financing is complex, posing challenges during emergencies. However, during the COVID-19 pandemic, the government eased regulations to accelerate funding and streamline the vaccination program, primarily from

general allocation funds (DAU) refocusing and revenue-sharing funds. While the program was funded by both national and regional budgets, the majority came from the national budget (APBN). Although limited, a reserve fund mechanism for crises was also available. Despite regulations ease, many regions faced bureaucratic barriers and challenges in absorbing the funds, along with limited funds to cover the pandemic response and vaccination efforts.

The central government gradually withdrew special COVID-19 financing amid competing priorities and limited resources. These regulatory changes for budget refocusing and reallocation between 2021 and 2022, also signaled a declining commitment and urgency to COVID-19 [48,57]. Consequently, numerous DHOs had to rely on their routine immunization budgets or allocate funds for COVID-19 vaccination that were not initially planned [53]. While the MoH outlines eligible components for national and regional funding, not all regions can cover every component due to varying capacities and needs.

This study found that, at the subnational level in Indonesia, the amount, type, and source of health expenditures vary significantly depending on fiscal capacity, political commitment, national and regional priorities, access challenges, geography, existing immunization infrastructure, and private sector engagement, among others. Districts often outspend provinces as provinces handle distribution and logistics, while DHOs incur additional costs for incentives, transportation, and other logistics for vaccination delivery.

At the provincial level, a relatively linear relationship exists: the higher the total COVID-19 vaccination expenditure per capita, the greater the vaccination coverage, especially in areas with limited private sector engagement. Provinces closer to Java tend to have greater fiscal capacity and achieve better vaccination coverage, while those farther away tend to have limited fiscal resources, therefore allocated less for COVID-19 vaccination efforts, limiting their pandemic response. Evidence shows that increased health expenditure improves health outcomes [58]. However, this pattern is less consistent at the district level. Except for cases like Gianyar, where private engagement is stronger, higher expenditures do not always translate to better coverage, indicating no clear linear trend at the district level. A comparison of Gianyar and Tanggamus highlights how local context shapes vaccination spending. Gianyar,as a big tourist destination and national priority area, received strong support from private sector and community partnerships, reallocating broader COVID-19 funds to support vaccination therefore no dedicated local budget. In contrast, Tanggamus, not being a national priority area, had to allocate and use primarily its own budget to cover COVID-19 vaccination program. These differences reflect variations in local capacity, national priority, governance, and stakeholder engagement influenced the prioritization and use of resources during the pandemic response. Furthermore, access to vaccines from the demand perspective is also related to economic, demographic, and social characteristics, as well as disability, location, and insurance coverage [59]. Individuals with higher levels of education, higher income, and those residing in Java or Bali received the first vaccine dose more promptly than those with lower educational backgrounds, lower income, or living in other regions [60].

The financial resources of PHOs primarily come from the regional budget (APBD), as provinces are mainly responsible for distributing resources to DHOs, requiring less funding compared to districts. Some PHOs extended funding to support vaccination delivery, improve cold chain equipment, and other activities. However, most district-level COVID-19 vaccination funding came from the national budget (APBN), via reallocations of BOK/DAK Non-Fisik and DID funds, as districts manage primary healthcare facilities. This implies a reliance on national funding for local vaccination programs.

Subnational governments in Indonesia shifted their COVID-19 vaccination budgets from medical equipment and materials to vaccination delivery and other health services between 2021 and 2022. While PHOs and DHOs prioritized public health and prevention, their focus was on operational needs such as honorariums, transportation, vaccine distribution, and cold chain equipment. However, not all regions allocated funds for vaccinators [9]. The MoH had not established specific policies on incentives for vaccinators, though honorariums are encouraged as part of the pandemic recovery strategy [53]. Ultimately, providing these incentives depended on each region's capacity and policies.

Most PHOs and DHOs allocated their expenditures exclusively for COVID-19 vaccination activities in 2021 and 2022, without integrating them into other healthcare programs. However, in Lampung, a significant portion of 2021 spending

went to integrated components due to the procurement of cold chain infrastructure, which supported both COVID-19 vaccination and other immunization efforts. Subnational governments made significant investments in cold chain facilities, addressing shortages, particularly for vaccines requiring specific temperatures [61].

This was followed by procurement of medical equipment, including Personal Protective Equipment (PPE), and supporting services like M&E. Some regions allocated more funds to support than to core components. For example, due to external support for its core vaccination activities, Gianyar allocated 100% of its budget to M&E, and Bali's 2022 budget focused solely on office supplies, rather than core vaccination activities. While monitoring, evaluation, and training is an integral part in public health emergencies [62], however, not all governments allocated budgets for M&E. Furthermore, we found that not all regions had budgets for COVID-19 vaccine distribution, particularly in areas with geographical challenges, leading to distribution delays [63]. These disproportionate allocations may stem from limited technical guidance and regulations, varying administrative capacities, and limited focus on accountability-related activities such as M&E. In some cases, core elements like vaccines or cold chain equipment were directly provided by the central government, leading subnational budgets to focus on complementary support. Additionally, logistical constraints in remote areas often required higher investment in coordination or training to enable delivery.

Although Indonesia's regional autonomy policy gives regions authority over local matters, the central government should provide technical guidelines for financing pandemic vaccination programs. These guidelines should assist the subnational governments to define expenditure proportions for core and support components across regions. "Core" refers to funding that goes directly to beneficiaries, such as vaccines distribution, cold chain equipment, and PPE. While "support" refers to activities that are supplemental to core functions, such as meetings and coordination before the intervention [64]. Furthermore, these guidelines should be formalized by regulations, particularly following the removal of health mandatory spending for the central and subnational governments under the Health Law starting 2023 [65]. Crucially, spending on support should not match or exceed that for core activities, ensuring proper focus and budget allocation that will promote ideal budgeting practices for each component.

There was no dedicated budget for COVID-19 vaccination programs or pandemic response targeting vulnerable groups in Indonesia, largely due to a lack of data on these populations. This study provides a portrayal of expenditure of COVID-19 vaccination programs for the Indonesian government in preparing for future crises. Improving the mechanism for budget transfers from the central to subnational levels, including consideration of vulnerabilities and geographical challenges, is essential to ensure that all regions have equal access to resources during a pandemic.

This study suggests the need for guidelines specifying budget and expenditure percentages, especially during health crises. Various technical guidelines and regulations for crisis funding management should be effectively utilized for regional budgeting in the COVID-19 vaccination program. The guidelines should account for regional vulnerabilities and capacity differences while allowing for flexibility. Understanding regional COVID-19 health burden, including infections, hospitalizations, and deaths, remains pivotal for effective allocation of resources and planning. However, the limited data available at the subnational level constrained a comprehensive assessment of system responsiveness. Within this constraint, case fatality rate (CFR) trends offer some insight. Following the 2021 vaccine rollout, CFRs over the 2020–2023 period generally declined in Central Sulawesi and Lampung, while Bali experienced a slight increase and Maluku showed fluctuations due to very low case and death numbers. However, confirmed cases and deaths dropped significantly across all provinces by 2023 [66–69]. While a direct correlation between spending and COVID-19 burden cannot be established, these findings provide preliminary insights into how variations in fiscal capacity, political commitment, national and regional priorities, access challenges, geography, existing immunization infrastructure, and private sector engagement, influenced pandemic response and vaccine fund allocation.

Building on these insights, a robust supervision and support mechanism is needed from central, provincial, to district/city governments, tailored to their varying planning capacities. This assistance will help subnational governments with more effective prioritization and resource allocation. Achieving accurate budgeting and effective spending requires

cross-sector collaboration and precise data [70]. Strengthening partnerships and involving external stakeholders, including CSOs and the private sector, could enhance the COVID-19 vaccination program's reach and impact.

While the study offers valuable insights specific to Indonesia, its generalizability to regions with different health systems and socio-economic conditions is limited. Nonetheless, by examining diverse regions within Indonesia with varying socio-economic profiles and policies, it highlights unique challenges and strengths that inform broader health system understanding and policy development in similar contexts.

### Strength and limitation

To our knowledge, this is the first study in Indonesia to track expenditures for the COVID-19 vaccination program involving PHOs and DHOs. Our study combines quantitative data with qualitative insights to provide a more comprehensive understanding. While the quantitative data is limited to PHO and DHO records and does not cover all COVID-19 financing sources in Indonesia, the qualitative data helped identify additional funding sources. However, we acknowledge that the qualitative approach relies on expert judgment, which carries inherent subjectivity and potential recall bias. Additionally, we excluded Jakarta and Papua due to data limitations. The exclusion of Papua may limit representation of provinces with significant geographical challenges, but similar contexts are captured through Maluku and Central Sulawesi. Although excluding Jakarta omits data from a province with high financing capacity and good regional access, Bali and Lampung serve as comparable proxies for financial capacity and ease of access within the study.

The lack of publicly available, disaggregated financial data on routine immunization prevented examination of the potential relationship between past investments in routine immunization and COVID-19 vaccine delivery. Provinces with stronger routine immunization systems may have been better positioned to implement COVID-19 vaccination effectively. Future research should explore these financial and infrastructural linkages to better understand the role of existing immunization systems in pandemic response.

### Conclusion

This study found that expenditure allocation for the COVID-19 vaccination program at the subnational level varied significantly. This is influenced by fiscal capacity, political commitment, national and regional priorities, access challenges, geography, existing immunization infrastructure, and private sector engagement, among others. The budget transfer mechanism from central government to subnational levels has not been distributed equally, causing disparities in vaccination coverage.

This study has broader relevance and can be a reference for the LMICs setting which had limited vaccine cold chain infrastructure and diverse archipelagic geography which faced similar challenges for decentralized vaccine programme delivery, especially during the emergency situation. Variations in budget commitment and spending can be minimised by having specific guidelines that prioritize core activities while allowing budget flexibility and funding mechanisms that can be mobilized to strengthen local governments capacity.

### Supporting information

**S1 Appendix. Operational definition of variables for cost classification (Source: RI – USAID, 2021) [41].**
(DOCX)

**S2 Appendix. Data collection matrix.**
(DOCX)

**S3 Appendix. In-depth interview participant.**
(DOCX)

**S4 Appendix. Focus group discussion participant.**
(DOCX)

**S5 Appendix. Budget and expenditure trends for the COVID-19 vaccination program.**
(DOCX)

**S6 Appendix. Expenditure per capita (US$) and vaccine coverage.**
(DOCX)

**S7 Appendix. Expenditure tracking.**
(DOCX)

## Acknowledgments

Authors sincerely thank Center for Indonesia's Strategic Development Initiatives (CISDI) and the vaccine coalition for vulnerable groups for supporting this study.

## Author contributions

**Conceptualization:** Olivia Herlinda, Ayudina Larasanti.

**Data curation:** Ayudina Larasanti, Sinta Amalia Kusumastuti Sumulyo.

**Formal analysis:** Olivia Herlinda, Ayudina Larasanti, Sinta Amalia Kusumastuti Sumulyo.

**Investigation:** Indra Yudha Mambea.

**Methodology:** Olivia Herlinda, Ayudina Larasanti, Sinta Amalia Kusumastuti Sumulyo, Indra Yudha Mambea.

**Project administration:** Ayudina Larasanti, Sinta Amalia Kusumastuti Sumulyo, Sayyid Muhammad Jundullah.

**Resources:** Ayudina Larasanti.

**Software:** Ayudina Larasanti.

**Supervision:** Olivia Herlinda.

**Validation:** Ayudina Larasanti, Sinta Amalia Kusumastuti Sumulyo, Sayyid Muhammad Jundullah.

**Visualization:** Ayudina Larasanti.

**Writing – original draft:** Olivia Herlinda, Ayudina Larasanti, Sinta Amalia Kusumastuti Sumulyo, Sayyid Muhammad Jundullah.

**Writing – review & editing:** Olivia Herlinda, Ayudina Larasanti, Sayyid Muhammad Jundullah, Indra Yudha Mambea.

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
