## [Decision Letter · Decision Letter 0]

27 Mar 2025

PGPH-D-24-02764

Tracking the Financial Flows of Indonesia's COVID-19 Vaccination Program

Dear Dr. Larasanti,

Thank you for submitting your manuscript to PLOS Global Public Health. After careful consideration, we feel that it has merit but does not fully meet PLOS Global Public Health’s publication criteria as it currently stands. Therefore, we invite you to submit a revised version of the manuscript that addresses the points raised during the review process.

We look forward to receiving your revised manuscript.

Kind regards,

Syed Shahid Abbas, MBBS, MPH, Ph.D.

Academic Editor

Additional Editor Comments (if provided):

Thank you for your patience with the time taken with the review process. Most reviewers found this to be an interesting research as demonstrated in their positive reviews. However, the manuscript requires much more clarity in methods, and discussion sections as highlighted by Reviewers 2 & 3. I look forward to reviewing the manuscript after you have had a chance to reflect on the comments and revise accordingly.

Reviewers' comments:

Reviewer's Responses to Questions

**Comments to the Author**

1. Does this manuscript meet PLOS Global Public Health’s publication criteria?

Reviewer #1: No

Reviewer #2: Yes

Reviewer #3: Partly

Reviewer #4: Yes

2. Has the statistical analysis been performed appropriately and rigorously?

Reviewer #1: N/A

Reviewer #2: Yes

Reviewer #3: No

Reviewer #4: N/A

3. Have the authors made all data underlying the findings in their manuscript fully available (please refer to the Data Availability Statement at the start of the manuscript PDF file)?

Reviewer #1: Yes

Reviewer #2: Yes

Reviewer #3: No

Reviewer #4: No

4. Is the manuscript presented in an intelligible fashion and written in standard English?

Reviewer #1: No

Reviewer #2: Yes

Reviewer #3: Yes

Reviewer #4: Yes

Reviewer #1: Lines 45-47, when you refer to coverage is it one dose, two doses? Please explain.

Line 79, there are many more studies that could be cited, e.g. https://doi.org/10.1016/j.vhri.2022.04.004.

Overall, the description does not have an orderly structure and is also very simplistic.

I don't think the study adds value to the literature.

Revise the English wording of the paper.

Reviewer #2: Introduction: Please specify the number of doses required for a complete vaccination in Indonesia. Additionally, clarify whether the 23% coverage in Papua refers to the primary dose or the complete vaccine series. Also, provide a definition of the vulnerable groups mentioned.

Methods: There is no explanation of how the authors selected the districts. It appears that Sigi had lower coverage than Central Sulawesi—please clarify this.

Methods: For better flow, consider moving the Study Population section above the explanation of the Quantitative and Qualitative Design.

Methods: What was the purpose of the qualitative study? Was it used for triangulation? If so, what aspects were being triangulated? Did it aim to explore challenges related to vaccination?

Figure 2: I suggest removing this figure and incorporating its findings into the text instead.

Figure 3: In 2021, Lampung spent $677,827, which was significantly higher than any other province. Given that Lampung's GDP was only twice that of Bali, how did Lampung manage such high expenditures? Were there additional financial resources involved?

Figures 10–11: These figures illustrate differences in cost allocations between provinces and districts, and the variations are striking. For example, Lampung and Maluku allocated most of their budget to equipment, while Bali and Sulawesi prioritized preventive services. Could it be that similar expenses were classified under different cost categories in the NHA/PHA? The same pattern is observed at the district level. Additionally, please provide more details about the cost categories to improve reader understanding—for instance, where were travel costs included?

Number of figures: There are many figures in the manuscript. Please consider merging some into a single figure with multiple panels.

Spending on routine vaccines: Did the authors analyze spending on routine vaccines at the provincial and district levels? This analysis could help explain why certain provinces invested more than others and had higher vaccine coverage. It is reasonable to assume that provinces or districts with higher routine vaccine coverage and spending would also have higher COVID-19 vaccine coverage. In the case of Lampung, which invested more in the cold chain, was this due to an already well-established routine vaccination infrastructure?

Funding allocation: Some regions allocated more funding to "Support" than to the "Core Component." Can the authors provide an explanation for this finding? Were the vaccines donated by entities other than the local government?

Lines 427–431: Was the spending correlated with the COVID-19 burden in each region? There is no explanation provided. It is important to assess how the system responded to the burden of the pandemic.

Reviewer #3: The manuscript provides an in-depth analysis of COVID-19 vaccine equity, health systems, and policy perspectives. The review of challenges and opportunities in vaccine distribution and access—particularly in Indonesia—is comprehensive, and the discussion on the economic evaluation of health interventions and the impact on health workers is timely. The extensive reference list demonstrates a robust engagement with current literature. However, the methodology section would benefit from additional detail regarding the costing processes and the interpretation of results. Although the Indonesia focus offers valuable context-specific insights, the generalizability of the findings to regions with differing health systems and socio-economic conditions is limited. Finally, further clarification is needed on how potential biases in data collection and analysis were addressed, and on the methods used to normalize the analysis, especially in light of the reliance on expert judgment due to data limitations.

Reviewer #4: This is an excellent paper that documents and adds to the body of knowledge on an important lesson from the Covid pandemic. The paper is well written and presents clear lessons for future pandemic preparedness. This will be vital for the global health community and should be shared widely. Some minor final edits as outlined below would be beneficial.

Methods:

The authors could provide further explanation for choosing the 4 PH and 4 DHOs. Were they purposively chosen because of their proximity to the capital for instance, what criteria was used for choosing a particular province out of all the provinces that met a particular criteria?

In the quantitative description, kindly explain how the expert judgement method was applied, to enable reproducibility.

Results/Discussion:

Were any reasons explored or found for the difference in spending between the provinces? For instance; Tanggamus and Gianyar.

In the limitation, kindly discuss any possible impact of exclusion of the two provinces without data on the study.

Conclusion

There are many important lessons from this study which are major for better pandemic preparedness, and the authors have highlighted these. The authors could further strengthen this section by reflecting further on how other countries can apply these lessons in their own settings.

**Do you want your identity to be public for this peer review?** For information about this choice, including consent withdrawal, please see our Privacy Policy

Reviewer #1: No

Reviewer #2: No

Reviewer #3: No

Reviewer #4: No

---

## [Decision Letter · Decision Letter 1]

24 Jun 2025

PGPH-D-24-02764R1

Tracking the Financial Flows of Indonesia's COVID-19 Vaccination Program

Dear Dr. Larasanti,

Thank you for submitting your manuscript to PLOS Global Public Health. After careful consideration, we feel that it has merit but does not fully meet PLOS Global Public Health’s publication criteria as it currently stands. Therefore, we invite you to submit a revised version of the manuscript that addresses the points raised during the review process.

We look forward to receiving your revised manuscript.

Kind regards,

Syed Shahid Abbas, MBBS, MPH, Ph.D.

Academic Editor

Journal Requirements:

Additional Editor Comments (if provided):

Reviewers' comments:

Reviewer's Responses to Questions

**Comments to the Author**

Reviewer #3: All comments have been addressed

Reviewer #4: All comments have been addressed

publication criteria?

Reviewer #3: Partly

Reviewer #4: Yes

3. Has the statistical analysis been performed appropriately and rigorously?

Reviewer #3: No

Reviewer #4: N/A

4. Have the authors made all data underlying the findings in their manuscript fully available (please refer to the Data Availability Statement at the start of the manuscript PDF file)?

Reviewer #3: Yes

Reviewer #4: Yes

5. Is the manuscript presented in an intelligible fashion and written in standard English?

Reviewer #3: Yes

Reviewer #4: Yes

Reviewer #3: The study offers critical insights into subnational COVID-19 vaccination financing in Indonesia over 2021 and 2022. However, it currently applies a fixed 2021 exchange rate (US$1 = IDR 14,308) to all expenditure data, without adjusting for exchange rate changes or domestic inflation between the two years. Given that vaccines and other key program inputs are predominantly imported, exchange rate depreciation and rising inflation in 2022 likely had a substantial impact on procurement costs and purchasing power at the subnational level. This could affect the accuracy of per capita expenditure comparisons and the interpretation of funding adequacy over time. The authors are encouraged to: (1) acknowledge this limitation in both the Methods and Discussion sections; and (2) where feasible, re-calculate or conduct a sensitivity analysis using the average 2022 exchange rate and incorporate inflationary/exchange rate adjustments to better reflect real-term expenditure changes.

Reviewer #4: The Authors have satisfactorily addressed comments and enriched the paper

**Do you want your identity to be public for this peer review?** For information about this choice, including consent withdrawal, please see our Privacy Policy

Reviewer #3: No

Reviewer #4: No

---

## [Editor Report · Decision Letter 2]

24 Jul 2025

Tracking the Financial Flows of Indonesia's COVID-19 Vaccination Program

PGPH-D-24-02764R2

Dear Ms Larasanti,

We are pleased to inform you that your manuscript 'Tracking the Financial Flows of Indonesia's COVID-19 Vaccination Program' has been provisionally accepted for publication in PLOS Global Public Health.

Best regards,

Syed Shahid Abbas, MBBS, MPH, Ph.D.

Academic Editor